# The Effect of Resistance Training on the Rehabilitation of Elderly Patients with Sarcopenia: A Meta-Analysis

**DOI:** 10.3390/ijerph192315491

**Published:** 2022-11-22

**Authors:** Haotian Zhao, Ruihong Cheng, Ge Song, Jin Teng, Siqin Shen, Xuancheng Fu, Yi Yan, Chang Liu

**Affiliations:** 1Department of Physical Education, Jiangnan University, Wuxi 214122, China; 2School of Sports Engineering, Beijing Sport University, Beijing 100084, China; 3School of Sport Science, Beijing Sport University, Beijing 100084, China; 4Faculty of Sports Science, Ningbo University, Ningbo 315211, China; 5Faculty of Engineering, University of Pannonia, 8200 Veszprém, Hungary

**Keywords:** sarcopenia, resistance training, optimal training parameters, elastic bands

## Abstract

Resistance training is considered to be an efficient treatment for age-related sarcopenia and can improve muscle strength and quality in patients. However, there are currently no recommendations on resistance training parameters to improve muscle strength and quality in elderly patients with sarcopenia. We conducted a systematic review and meta-analysis of randomized controlled trials (RCTs) and included 13 eligible RCTs. Resistance training significantly improved grip strength, gait speed, and skeletal muscle index in patients with age-related sarcopenia, and kettlebell was found to be the most effective modality. However, it is noteworthy that the elastic band is also a recommended form of resistance training considering that the kettlebell intervention was tested in only one study, while the elastic band was confirmed by multiple studies. Elastic band training (Hedges’s g = 0.629, 95%CI = 0.090–1.168, *p* < 0.05) (40–60 min per session, more than three times per week for at least 12 weeks) was the most efficient training method. Thus, resistance training can significantly improve muscle strength and muscle quality in elderly patients with sarcopenia. In addition, moderate-intensity resistance training using elastic bands may be the best training prescription for elderly patients with sarcopenia.

## 1. Introduction

Skeletal muscle quality and function decline with age, and previous studies have shown that skeletal muscle mass will decrease at a rate of 1–2% per year after the age of 50, and skeletal muscle strength will reduce by 1.5% at age 50–60, even leading to sarcopenia [1]. Sarcopenia, also known as muscle atrophy [2], is a persistent disease in modern geriatrics and is a progressive, systemic decrease in muscle mass, muscle strength, or physiological muscle function associated with aging. Typically, the prevalence of sarcopenia also increases with age [3]. Sarcopenic individuals are highly susceptible to falls and disability, and there is an inflection point for end-stage functional decline [4].

In the elderly, the etiology of sarcopenia is multifactorial. Under normal conditions, physiological muscle mass and function are related to a dynamic balance between positive and negative regulators of muscle growth. The central nervous, immune and endocrine systems coordinate this balance. The development and progression of sarcopenia depends on multiple mechanisms associated with the aging process, and the combination of these mechanisms can also lead to the disruption of normal skeletal muscle physiological function [5].

Among the age-related changes in body composition that occur at the structural level of muscle fibers, changes in contractile properties and abnormalities in neuromuscular connectivity are particularly evident [6]. Muscle atrophy and dysautonomia due to sarcopenia also involve several pathogenic mechanisms related to hormonal status, inflammatory status, insulin resistance, telomere shortening, and oxidative stress [7].

Muscle strength and resistance in older adults is important for the health of older adults and is a topical issue in public health [8]. Decreased muscle mass is one of the main causes of skeletal damage in older adults, and growing evidence also suggests that the loss of muscle mass in older adults is strongly associated with frailty, which in turn is strongly associated with function and disability [9], in addition to contributing to problems such as type 2 diabetes, hypertension, and obesity, secondary conditions that also place a heavy burden on the health care system [10].

Current studies have found that [11,12] resistance training can alleviate age-related skeletal muscle degeneration, improve muscle strength, regulate blood glucose, and control blood pressure in the elderly. However, the training program for resistance training of skeletal sarcopenia in the elderly has not been standardized [13]. Additionally, there are still controversies about the rehabilitation training plan of resistance training for the elderly with sarcopenia. The number of subjects in each study is limited, the research scope is limited, the reliability of a single experimental study is low, and the evaluation of the resistance training effect lacks objectivity. Therefore, the integration and analysis of previous studies is essential to improve the objectivity of the evaluation of the effects of resistance training.

Previously, many similar meta-analyses were conducted by distinguished researchers [11,13], showing the high prevalence of sarcopenia in most older adults and the need for well-designed standardized studies to evaluate exercise before developing guidelines for the treatment of the elderly with sarcopenia [14]. Physicians should screen for sarcopenia in the community and in geriatric homes and make a diagnosis based on muscle mass and function. Supervised resistance training is recommended for individuals with sarcopenia [12]. A resistance training program lasting ≥8 weeks should be considered a highly effective preventive strategy for delaying and mitigating the negative effects of early and late onset sarcopenia and frailty in individuals ≥65 years of age diagnosed with pre-muscular dystrophy, sarcopenia, pre-frailty, or frailty [13]. Although all of these studies showed resistance exercise to be highly beneficial for older adults with sarcopenia, the specific gender of the subjects, the intervention period, the frequency of the intervention, the duration of the single intervention, and the type of resistance of the included population have not been discussed.

Given that the reference value of existing studies is not high and cannot be applied to the elderly, resistance training was chosen as the starting point of this study. From the perspective of evidence-based medicine, the research data on the effect of resistance training on the rehabilitation of elderly patients with sarcopenia were summarized for meta-analysis. In addition, this study explored the effects of different intervention subjects, intervention time, intervention methods, and other factors on outcome indicators through subgroup analysis. This study provides a reliable theoretical reference for the rational selection of resistance training to prevent the related diseases caused by sarcopenia in the elderly.

## 2. Materials and Methods

### 2.1. Literature Search

The literature search followed the PRISMA (Preferred Reporting Items for Systematic Reviews and Meta-Analyses) guidelines. Comprehensive literature searches were conducted on five online databases (CNKI (China National Knowledge Infrastructure), Wan fang Data, PubMed, Web of Science, and EBSCO), and the search period was up to August 2022. The search was performed using Boolean operations in combination with the following terms: “elderly”, “older”, “sarcopenia”, “sarcopenias”, “resistance training”, “strength training”, and “randomized controlled trial”. Additional searches were conducted within the reference lists of the included records.

### 2.2. Inclusion Criteria

Inclusion criteria: (1) subjects: middle-aged and older adults aged 60–91; (2) study type: randomized controlled trial (RCT); (3) interventions: resistance training; (4) the study subjects met the diagnostic criteria for sarcopenia stipulated by the European Working Group on skeletal sarcopenia in the elderly (EWGSOP) and the Asian Working Group on skeletal sarcopenia (AWGS); (5) outcome measures included at least one of the following factors: grip strength, gait speed, and skeletal muscle index.

### 2.3. Study Selection and Data Extraction

The Endnote citation management software was applied to conduct the literature screening. The extracted information from the literature included: the first author of the paper, the year of publication, the sample-size number, the characteristics of the participants (gender, the number of male and female participants, age), the elements of the intervention (training modes, intervention protocol, duration, frequency), and the outcomes. Literature screening and data extraction were conducted independently by two researchers (HZ and JT), who consulted each other after completion. If any disagreement occurs, the study was referred to a third researcher (SS) for further discussion and decision.

### 2.4. Assessment of Methodological Quality and Statistical Analysis

The methodological quality of the included studies was assessed using the Physiotherapy Evidence Database (PEDro) scale, which involves 11 items (eligibility criteria, randomized assignation, concealment of allocation, homogeneity between groups, blinded subjects, blinded trainers, blinded testers, dropout rate less than 15%, intention-to-treat analysis, between-group comparisons, and variability measurements) [15]. The items were rated as 1 or 0 (+ or −) depending on whether the item in the study included explicit information. When the total score of all items was greater than or equal to six, the study was considered to have a high study quality, while scores of 4–5 and less than 4 corresponded to moderate- and poor-quality studies, respectively [15,16].

### 2.5. Statistical Analysis

Mean Difference (MD) and 95% Confidence Intervals (CI) were calculated between the intervention group and the control group after the intervention to quantify the effect of resistance training on indicators related to elderly patients with sarcopenia. If different studies used different measurement methods and tools, Hedges’s g and 95%CI were used to measure the effect size. The effect size was divided into three categories: small (<0.5), medium (0.5–0.8) and large (>0.8). Collective effects were calculated using a random effects model. After calculating the summary effect, taking grip strength as the entry point, subgroup analysis (gender, intervention method, intervention cycle, intervention frequency, single intervention duration, age), and inter-subgroup comparisons were conducted to determine the best resistance training parameters to improve sarcopenia. Funnel plots of more than ten studies were visually evaluated to determine the risk of publication bias on the results. Univariate or multivariate adjustment analysis was used to assess the impact of potential covariates on meta-analysis. All analyses were statistically analyzed using Stata 17.0 software (StataCorp LP, College Station, TX, USA).

## 3. Results

### 3.1. Literature Characteristics

Figure 1 presents the detailed search results for each stage. A total of 723 studies were initially retrieved, and finally, only 13 studies and 569 subjects were included in this meta-analysis.

In this meta-analysis, grip strength, gait speed, and skeletal muscle index were mainly used to evaluate the intervention effect of resistance training. Eleven studies included grip strength, eight included gait speed, and six included skeletal muscle index. All included studies were randomized controlled trials, as shown in Table 1.

### 3.2. Methodological Quality of the Included Studies

According to the PEDro scale, 12 included studies were of high quality, with scores of 6–11, 2 studies were of moderate quality, with scores of 4–5, and no low-quality studies were included (Table 2).

### 3.3. Effect of Resistance Training on Grip Strength, Gait Speed, and Skeletal Muscle Index

The effects of resistance training on both grip strength and gait speed showed a high degree of heterogeneity (I^2^ = 59.37%, *p* < 0.05 and I^2^ = 93.10%, *p* < 0.05), while the effects of resistance training on skeletal muscle index showed good inter-study heterogeneity (I^2^ = 2.70%, *p* = 0.40).

In this study, a random effects model was conducted for the three subjects mentioned above. Combining the effect quantities, the results showed that Hedges’s g = 0.60, 95%CI = 0.30–0.89 (*p* < 0.05), for the effect of resistance training on grip strength; Hedges’s g = 1.50, 95%CI = 0.59–2.40 (*p* < 0.05,) for the effect of resistance training on gait speed; and Hedges’s g = 0.52, 95%CI = 0.27–0.76 (*p* < 0.05), for the effect of resistance training on skeletal muscle index. These results suggest that resistance training significantly improves grip strength, gait speed, and skeletal muscle index in elderly patients with sarcopenia (Figure 2).

### 3.4. Results of Subgroup Analysis

#### 3.4.1. Gender

In the comparison of subgroup analyses, we did not find any significant differences in grip strength levels after resistance training in older patients with sarcopenia by gender (all tests between subgroups showed *p* < 0.05). Among them, the largest effect was found in the group where gender was not given (Hedges’s g = 0.762, 95%CI = 0.269–1.255, *p* < 0.01), followed by the female group (Hedges’s g = 0.642, 95%CI = 0.019–1.265, *p* < 0.05), and the mixed-gender group (Hedges’s g = 0.575, 95%CI = 0.257–0.893, *p* < 0.01) (Table 3).

#### 3.4.2. Intervention Methods

In the comparison of subgroup analysis, we found that there were significant differences in grip strength levels between intervention methods in elderly sarcopenia patients (*p* < 0.05 for the instrument group and kettlebell group; *p* < 0.01 for the elastic band group; *p* > 0.05 for the mixed group and the self-weight group). Kettlebell group had the largest effect size (Hedges’s g = 1.04, 95%CI = 0.317–1.763, *p* < 0.05) among them, followed by instrument group (Hedges’s g = 0.669, 95%CI = 0.0.051–1.287, *p* < 0.05), elastic band group (Hedges’s g = 0.629, 95%CI = 0.090–1.168, *p* < 0.05), mixed group (Hedges’s g = 0.508, 95%CI = −0.258–1.274, *p* = 1.274) and self-weight group (Hedges’s g = 0.448, 95%CI = −0.257–1.154, *p* = 0.213) (Table 3).

#### 3.4.3. Training Cycle

In the comparison of subgroup analysis, we found no significant difference in the grip strength level of elderly sarcopenia patients in different intervention cycles (*p* < 0.05 for all tests between subgroups; *p* < 0.01 for over 12 weeks’ group). Among them, the intervention cycle effect of the group over 12 weeks was the largest (Hedges’ g = 0.638, 95%CI = 0.252–1.025, *p* < 0.05), and that of the group below 12 weeks showed Hedges’ g = 0.541, 95%CI = 0.030–1.053, *p* < 0.05 (Table 3).

#### 3.4.4. Intervention Frequency

In the comparison of subgroup analyses, we found no significant difference in the grip strength level of elderly sarcopenia patients with different intervention frequencies (*p* < 0.05 for all tests between subgroups; *p* < 0.01 for over three times per week group). Among them, the effect of the over three times per week group was the largest (Hedges’s g = 0.927, 95%CI = 0.292–1.562, *p* < 0.01), and that of the below three times per week group showed Hedges’s’ g = 0.431(95%CI = 0.094–0.768, *p* < 0.05) (Table 3).

#### 3.4.5. Duration of Each Session

In the comparison of subgroup analyses, we found no significant difference in the grip strength level of elderly sarcopenia patients with different single intervention durations (*p* < 0.05 for over 60 min; *p* < 0.001 for the 40–60 min group). Among them, the single intervention duration of 40–60 min had the most significant effect (Hedge’s g = 0.666, 95%CI = 0.381–0.951, *p* < 0.01), and Hedges’s g = 0.596 (95%CI = 0.081–1.111, *p* < 0.05) for the group of over 60 min (Table 3).

#### 3.4.6. Age

In the comparison of subgroup analyses, we did not find a significant difference in the grip strength levels of elderly sarcopenia patients of different ages after resistance training *p* < 0.01 for the over 70 years old group, *p* < 0.001 for the below 70 years old group). The effect of the below 70 years old group was the largest (Hedges’s g = 0.719, 95%CI = 0.336–1.102, *p* < 0.01) (Table 3).

### 3.5. Publication Bias

Funnel plots were generated to determine the risk of publication bias for outcomes reported from more than ten studies (Figure 3). Noticeable asymmetries were found in those funnel plots, indicating obvious publication biases regarding proactive and static steady state equilibrium.

## 4. Discussion

This meta-analysis investigated the effect of resistance training on elderly patients with sarcopenia. This study found that: (1) resistance training showed a high degree of heterogeneity in the effects on grip strength and gait speed, while good between-study heterogeneity was demonstrated for skeletal muscle indicators; (2) resistance training significantly improved skeletal muscle index, grip strength, and gait speed in elderly patients with sarcopenia; (3) subgroup analysis indicated that different intervention methods had different effects on grip strength in elderly sarcopenia patients, and the elastic band was the best intervention method. From the perspective of effect quantity, the intervention cycle should be less than 12 weeks, the intervention frequency should not be less than three times/week, and the duration of a single intervention should not exceed 60 min. For elderly sarcopenia patients over 70 years old, the effect of resistance training is not obvious. In addition, subgroup analysis found that the heterogeneity between the studies of resistance training on grip strength may be related to the gender of the intervention objects. Studies by Rena [30] and others have proved significant differences in interleukin-6 and C-reactive protein levels between genders.

### 4.1. Effect of Resistance Training on Grip Strength

As a low-cost and effective treatment, resistance training has gradually been shown to be effective in the rehabilitation of sarcopenia. Previous studies have shown that resistance training can stimulate the proliferation of skeletal muscle satellite cells, improve the rate of protein synthesis, enhance the strength of skeletal muscle, improve the quality of skeletal muscle and neuromuscular adaptability, and prevent or delay the occurrence of sarcopenia [31].

Grip strength is a simple and effective method to evaluate upper-limb strength, which has been widely used in the clinic to evaluate the upper-limb strength of patients with sarcopenia [20]. Previous studies have shown that grip strength can effectively predict the decline of skeletal muscle mass. Grip strength is significantly related to the risk of falls, hospitalization, and mortality in the elderly [32]. This meta-analysis showed that resistance training significantly improves the grip and upper-limb strength of elderly sarcopenia patients. However, there was a high degree of heterogeneity among the included studies, which may be because the subjects in the included studies come from different regions and cultures. Some studies [33,34] have found differences in muscle mass measurement results among the elderly of different cultures. Even if the culture is the same, the measurement results will be biased due to the different geographical locations, which may reflect the influence of lifestyle factors on the development of sarcopenia.

### 4.2. Effect of Resistance Training on Gait Speed

Gait speed is an indicator to evaluate the skeletal muscle function of the lower limbs of patients with sarcopenia [35]. Several studies [36] have confirmed that resistance training improves the skeletal muscle function of lower limbs and gait speed. Villareal et al. [37] found that resistance training significantly improved gait speed in the elderly population, which is consistent with the results of our present study. However, there is considerable heterogeneity among gait speed studies included in this study, and the impact of resistance training on the gait speed in elderly sarcopenia patients still needs more experimental validation.

### 4.3. Effect of Resistance Training on Skeletal Muscle Mass Index

The skeletal muscle mass index is the square of the limbs’ skeletal muscle divided by the individual’s height. When the individual is two standard deviations lower than the reference value, the patient is determined to have sarcopenia [36]. This study found that resistance training significantly improved the skeletal muscle mass index of patients with sarcopenia. Vikberg et al. [38] found that ten weeks of resistance training significantly improved the patients’ skeletal muscle mass index. The research results of Liao et al. [22] reached a similar conclusion, which is consistent with the results of our present study.

### 4.4. Optimal Parameters of Resistance Training for Rehabilitation

The international guidelines for clinical practice of sarcopenia list physical activity as the highest level of rehabilitation evidence and point out that patients with sarcopenia should strengthen physical training based on resistance training. The guidelines emphasize that the best treatment scheme for sarcopenia is physical training combined with resistance training in a progressive manner [39]. A previous review [12] showed that resistance training more than three times a week might improve the muscle strength and quality of elderly sarcopenia patients. Our study further confirms this idea and indicates that resistance training for no more than 60 min per week significantly improves muscle strength and quality in elderly sarcopenia patients.

Regarding training frequency and duration, the current international clinical practice guidelines for sarcopenia did not detail the best recommendations for elderly sarcopenia patients [39]. The results of this subgroup analysis found that the effect of resistance training on elderly sarcopenia patients increased with increased training frequency.

In terms of training duration, we found that more than 12 weeks of resistance training had a more noticeable effect on improving the grip strength in elderly sarcopenia patients, which is consistent with the results of Monteiro [40]. However, the longest intervention time of the included studies was 24 weeks [26], so this result needs to be interpreted carefully.

There are various forms of resistance training, and subgroup analysis found kettlebells to be the most effective. However, it is worth noting that considering that there was only one trial of this intervention, whereas the elastic band was corroborated by multiple experiments, we believe that resistance training with the elastic band should also be a recommended form of resistance training. Furthermore, Liu Yiyi [41] concluded that resistance training with an elastic band can improve muscle strength and the quality of life of the elderly, which is consistent with the results of our paper. However, this study’s equipment and kettlebell training are limited, so this result may still need further research.

## 5. Conclusions

In conclusion, this meta-analysis found that resistance training can significantly improve muscle strength and muscle quality in elderly sarcopenia patients. Furthermore, moderate-intensity resistance training in the form of an elastic band may be the best training prescription for elderly sarcopenia patients, and training for more than 12 weeks, over three times a week, and 40–60 min each time were recommended.

## 6. Limitations

Due to the lack of detailed research on training intensity, we cannot determine the optimal training intensity parameters for resistance training. At the same time, due to the limited number of included studies, too few eligible studies were included in some subgroup analyses to draw reliable conclusions or even to use, resulting in significant heterogeneity between subgroups. In addition, although subgroup analysis was performed to identify possible sources of heterogeneity, the apparent heterogeneity among the included studies may affect the accuracy of the pooled results, thus limiting its generalizability. Future research should explore the relationship between other unique variables and resistance training. Due to these limitations, our results may only be preliminary, and more high-quality clinical trials are needed for validation.

## Figures and Tables

**Figure 1 ijerph-19-15491-f001:**
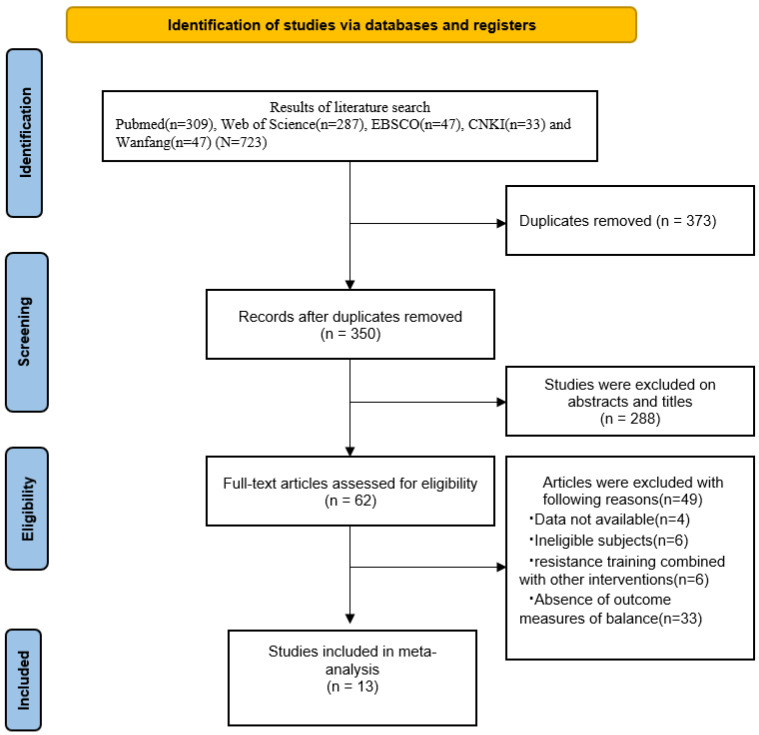
Literature selection process.

**Figure 2 ijerph-19-15491-f002:**
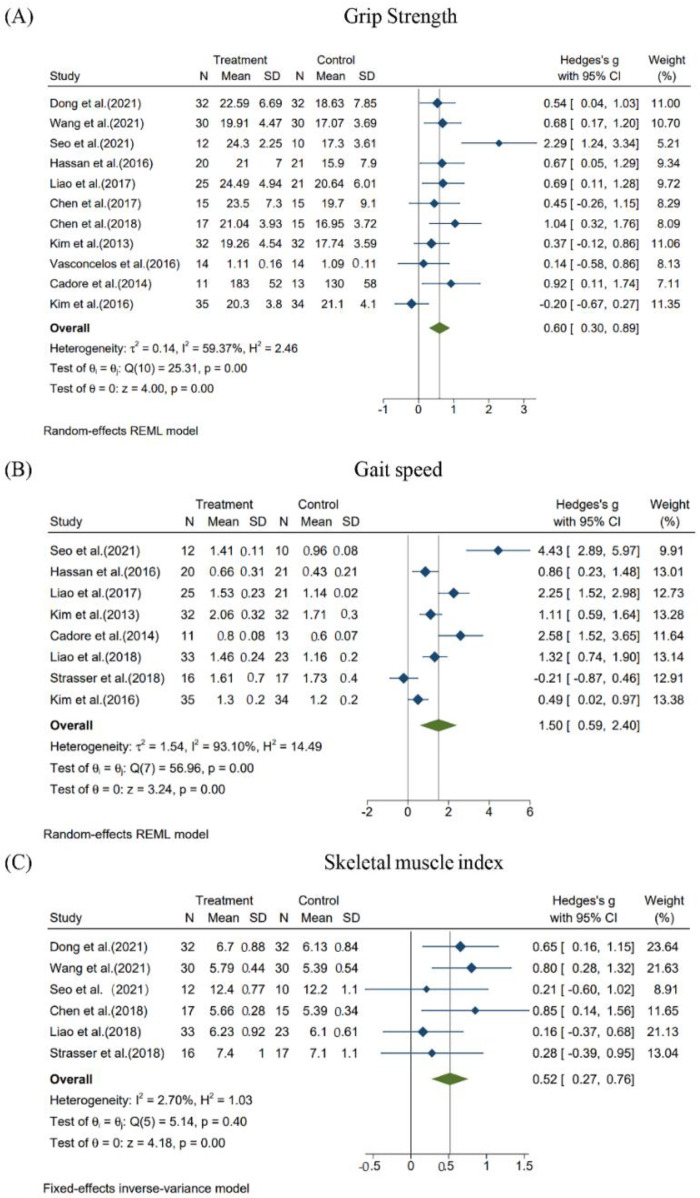
Effects of resistance training on grip strength, gait speed and skeletal muscle index in elderly patients with sarcopenia. (**A**) Effect of resistance training on grip strength. (**B**) Effect of resistance training on gait speed. (**C**) Effect of resistance training on skeletal muscle index [17,18,19,20,21,22,23,24,25,26,27,28,29].

**Figure 3 ijerph-19-15491-f003:**
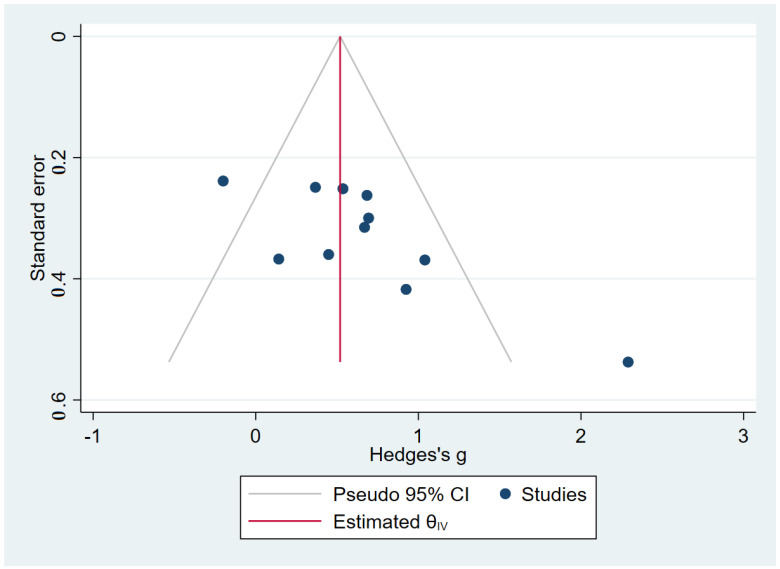
Publication bias funnel chart of the effect of resistance training on grip strength.

**Table 1 ijerph-19-15491-t001:** General features of selected research literature.

Author	Gender	Sample Size		Age		Intervention	Intensity	Frequency	Cycle	Duration (min)	Outcomes
		Experimental	Control	Experimental	Control						
Kim et al. (2013) [17]	Females	32	32	79.6 ± 4.2	80.2 ± 5.6	Elastic band (whole body)	2 sets; 10 times per set	2/week	12 weeks	60	Gait speed and grip strength
Cadore et al. (2014) [18]	NG	11	13	91.9 ± 4.1		Mixed type (whole body)	40–60%1 rm; 8–10 times per set	2/week	12 weeks	40	Gait speed and grip strength
Vasconcelos et al. (2016) [19]	Females	14	14	72 ± 4.6	72 ± 3.6	Mixed type (whole body)	50–75%1 rm	2/week	10 weeks	60	Grip strength
Kim et al. (2016) [20]	Females	35	34	81.4 ± 4.3	81.1 ± 5.1	Elastic band (upper body)	Moderate intensity; 3 sets; 10 times per set	2/week	12 weeks	60	Gait speed and grip strength
Hassan et al. (2016) [21]	NG	20	21	85.9 ± 7.5		Equipment training (whole body)	3 sets; 10–15 times per set	2/week	6 months	60	Grip strength
Liao et al. (2017) [22]	Females	25	21	66.39 ± 4.49	68.42 ± 5.86	Elastic band (whole body)	Moderate intensity	3/week	12 weeks	40	Gait speed and grip strength
Chen et al. (2017) [23]	Males, Females	15	15	68.9 ± 4.4	68.6 ± 3.1	Self-weight training(whole body)	60–70%1 rm; 3 sets; 8–12 times per set	2/week	8 weeks	60	Grip strength
Chen et al. (2018) [24]	Females	17	15	66.7 ± 5.3	68.3 ± 2.8	Kettlebell training (whole body)	60–70%1 rm; 3 sets; 8–12 times per set	2/week	8 weeks	60	Grip strength, skeletal muscle index
Liao et al. (2018) [25]	Females	33	23	66.67 ± 4.54	68.32 ± 6.05	Elastic band (whole body)	Moderate intensity; 3 sets; 10 times per set	3/week	12 weeks	40	Gait speed, skeletal muscle index
Strasser et al. (2018) [26]	Males, Females	16	17	82.0 ± 5.4	83.9 ± 5.8	Elastic band (upper extremity)	Moderate intensity	2/week	24 weeks	40	Gait speed, skeletal muscle index
Dong et al. (2021) [27]	Males, Females	32	32	60–85		Elastic band (whole body)	Moderate intensity; 3 sets; 12–15 times per set	3/week	12 week	40–50	Grip strength, skeletal muscle index
Wang et al. (2021) [28]	Males, Females	30	30	70.4 ± 5.1	70.7 ± 3.9	Elastic band (whole body)	50–70%1 rm; 3 sets; 8–12 times per set	2–3/week	12 weeks	45	grip strength, skeletal muscle index
Seo et al. (2021) [29]	Females	12	10	70.3 ± 5.38	72.9 ± 4.75	Elastic band (whole body)	8–15 rm; 3–5 sets; 6–15 times per set	3/week	16 weeks	60	Gait speed and grip strength, skeletal muscle index

**Table 2 ijerph-19-15491-t002:** Physiotherapy Evidence Database (PEDro) score of the included studies.

References	(1)	(2)	(3)	(4)	(5)	(6)	(7)	(8)	(9)	(10)	(11)	(12)
Kim et al. (2013) [17]	+	+	+	+	+	+	+	+	-	+	+	10
Cadore et al. (2014) [18]	+	+	-	-	-	-	+	+	-	+	+	6
Vasconcelos et al. (2016) [19]	+	+	-	-	-	-	-	+	+	+	+	6
Kim et al. (2016) [20]	+	+	-	+	-	-	-	+	+	+	+	7
Hassan et al. (2016) [21]	+	+	-	+	-	+	-	+	-	+	+	7
Liao et al. (2017) [22]	+	+	-	+	-	-	-	+	-	+	+	6
Chen et al. (2017) [23]	+	+	-	-	-	-	-	+	-	+	+	5
Chen et al. (2018) [24]	+	+	-	+	-	-	-	+	-	+	+	6
Liao et al. (2018) [25]	+	+	+	+	-	+	-	+	-	+	+	8
Strasser et al. (2018) [26]	+	+	-	+	-	-	-	+	-	+	+	6
Dong et al. (2021) [27]	-	+	-	-	-	-	-	+	+	+	+	5
Wang et al. (2021) [28]	+	+	-	-	-	-	-	+	+	+	+	6
Seo et al. (2021) [29]	+	+	+	+	-	+	-	-	-	+	+	7

Note: + indicates one point; - indicates no point. The score of the first-item eligibility criteria is not included in the total score. (1) Eligibility criteria; (2) randomized assignation; (3) concealment of allocation; (4) homogeneity between groups; (5) blinded subjects; (6) blinded trainers; (7) blinded testers; (8) dropout rate < 15%; (9) intention-to-treat; (10) between-group comparisons; (11) point and variability measurement; (12) total scores.

**Table 3 ijerph-19-15491-t003:** Subgroup analysis of the effect of resistance training on grip strength of elderly patients with sarcopenia.

Groups		N	Hedges’s g	95%CI		*p*-Value	I^2^%
Gender	Female	6	0.642	0.019	1.265	0.043	82.99
	Gender unknown	2	0.762	0.269	1.255	0.002	0.00
	Gender mixed	3	0.575	0.257	0.893	0.000	0.00
Intervention methods	Instrument	1	0.669	0.051	1.287	0.034	-
	Kettlebell	1	1.04	0.317	1.763	0.005	-
	Elastic band	6	0.629	0.090	1.168	0.022	82.11
	Mixed type	2	0.508	−0.258	1.274	1.274	49.56
	Self-weight	1	0.448	−0.257	1.154	0.213	-
Intervention cycle	≥12 weeks	8	0.638	0.252	1.025	0.001	70.90
	<12 weeks	3	0.541	0.030	1.053	0.038	34.73
Intervention frequency	≥3 times/week	4	0.927	0.292	1.562	0.004	76.47
	<3 times/week	7	0.431	0.094	0.768	0.012	50.19
Intervention duration (min)	≥60	7	0.596	0.081	1.111	0.023	77.68
	40–60	4	0.666	0.381	0.951	0.000	0.00
Age (years)	≥70	8	0.581	0.174	0.988	0.005	72.49
	<70	3	0.719	0.336	1.102	0.000	0.00
Overall		11	0.598	0.305	0.891	0.000	59.37

## Data Availability

Not applicable.

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
