# Peer review of "The Effect of Resistance Training on the Rehabilitation of Elderly Patients with Sarcopenia: A Meta-Analysis"

_ijerph, 2022, doi:10.3390/ijerph192315491_

Round 1

Reviewer 1 Report

An interesting paper - I have the following comments:

1. The abstract does not adequately summarise the whole manuscript.

2. The authors need to better highlight what additional information this manuscript adds to the topic, compared to the numerous other meta analyses that have been published over the last 5-10 years. At the moment, there is only one short brief statement where this main point is mentioned and it is as follows "....the gender, intervention cycle, intervention frequency, single intervention duration, and resistance type of the included population have not been discussed.....". However, this is far too brief and should be expanded into a whole paragraph, I think.

In relation to the methodology section, who were the persons that selected and extracted the relevant studies ? Please give us more information. Were they members of the author team, or someone independent ?

In the statistical analysis, please state what is meant by "MD and 95% CI" which were calculated between the intervention group and the control group.  Is MD "Mean Difference" ? CI = "Confidence Interval" ?

In the results section, (Effect of resistance training on......) the statement "There was high heterogeneity among studies" is simply repeated and does not give the reader much information. Perhaps the 3 sections here can be merged ?

Line 156: What does this mean " the effect size of the 156 not given genders group is the largest" ?

Line 160: I don't know what the following means (please explain this better)..."This section may be divided by subheadings. It should provide a concise and precise description of the experimental results, their interpretation, as well as the experimental conclusions that can be drawn" ?

In the abstract you say "moderate-intensity resistance training using 23 elastic bands may be the best training prescription for elderly sarcopenia patients." and yet, in the results, the largest effect size is for using the Kettleball (lines 167-168). As mentioned previously, the abstract does not adequately summarise the whole manuscript.

At the start of the discussion, there are the words "Authors should discuss the results and how they can be interpreted from the perspective of previous studies and of the working hypotheses. The findings and their implications should be discussed in the broadest context possible. Future research directions may also be highlighted". Please remove.

Line 214: Please remove the word "comprehensively".

In the section beginning on line 215: Please add something about the heterogeneity of studies that were included (mentioned in lines 134-146).

Line 228: Change the word "proven" for "shown to be".

Line 229: Change "intervention" for "rehabilitation".

Line 239:Change "could significantly improve" for "significantly improves".

Line 241: Remove the word "races" and replace with "cultures".

Line 243: Remove the word "races" and replace with "cultures".

Line 243: Remove the words "Even if the race is the same..." and replace with "Even if the cultural group is the same...".

Line 243: Can you better explain what you mean by "the measurement results will be biased due to different geographical locations" ?

Line 284: As mentioned earlier, your statement "we believe that resistance training with 284 the elastic band has the best effect..." is nor supported by your effect sizes (your meta analysis). So, it is not agreeable to make your statement here. It sounds like your are basing this statement on 'personal opinion'. So, you should rethink this and reword it here.

Author Response

Thanks for the reviewer’s professional comments.

Reviewer 2 Report

The authors performed an extensive investigation on resistance training in people with sarcopenia, but essential issues must be addressed:

The abstract should exclude headings.

Line 31- Please check reference 1

Line 32- "Sarcopenia, also called sarcopenia"??? please correct.

The authors should also point out other biological alterations of sarcopenia (like doi: 10.3389/fmed.2019.00184), and emphasise the importance of muscle strength and resistance in the elderly as a public health issue, healthcare system costs for secondary illnesses (risk of falls- fractures; obesity, etc.)  

The authors should provide a PRISMA checklist file for systematic reviews and a registration number for systematic reviews ( if possible).

The authors should describe, at least minimally, what kind of therapy the subjects within the studies performed. How was the exercise intensity set for the whole body, lower or upper extremity, and cardio-respiratory training?

Please check lines 210-213- from the Discussion section.

Author Response

(The authors gave the same response as above.)

Round 2

Reviewer 1 Report

Excellent revisions - thank you.

Reviewer 2 Report

The paper has significant improvements and is suitable for publication.